# Influence of Heifer Post-Weaning Voluntary Feed Intake Classification on Lifetime Productivity in Black Angus Beef Females

**DOI:** 10.3390/ani12131687

**Published:** 2022-06-30

**Authors:** Krista R. Wellnitz, Cory T. Parsons, Julia M. Dafoe, Darrin L. Boss, Samuel A. Wyffels, Timothy DelCurto, Megan L. Van Emon

**Affiliations:** 1Department of Animal and Range Sciences, Montana State University, Bozeman, MT 59717, USA; krista.eiseman@student.montana.edu (K.R.W.); samwyffels@montana.edu (S.A.W.); timothy.delcurto@montana.edu (T.D.); 2Northern Agricultural Research Center, Montana State University, Havre, MT 59501, USA; cory.parsons2@chsinc.com (C.T.P.); julia.dafoe@montana.edu (J.M.D.); dboss@montana.edu (D.L.B.)

**Keywords:** beef cattle, efficiency, heifer, intake, productivity

## Abstract

**Simple Summary:**

Replacement beef heifers represent the future of a herd for many cow-calf producers. Producers must invest significant resources into the development of these females as well as consider their maintenance and production costs as mature cows. As a result, selection focuses on replacement heifers that are more efficient in limited nutrition forage base systems. Producers can improve lifetime productivity by ensuring their replacements are of an ideal age and size to be bred as a 2-yr old. Selection criteria may include reproductive ability and breeding history, milk production, and weaning/yearling weights. The ability to select replacement females that are more efficient without negatively affecting overall productivity will help to decreased inputs and alleviate the need for additional expenses such as supplemental feed resources.

**Abstract:**

This study evaluated heifer post-weaning voluntary feed intake (g/kg BW) classification on performance and reproductive measures, as well as impacts on lifetime productivity of 519 commercial Angus beef females. Heifer post-weaning voluntary feed intake (g/kg BW) was calculated over 80 test days following weaning using GrowSafe units. Heifers were categorized based on voluntary feed intake (g/kg BW) as either low (<−0.50 SD from the mean), average (±0.50 SD from the mean), or high (>0.50 SD from the mean) within year. Cow body weight (BW) and body condition score (BCS) at breeding displayed an age effect (*p* < 0.001), with 2- and 3-year-old cows having lighter BW and lower BCS than 4-yr-old and older cows. Cow BW at weaning showed significance for age and intake (*p* < 0.001) with younger cows being lighter than older cows, while low intake classified females had greater BW at weaning compared to average and high intake females. Additionally, calf 205-d weights and calf weaning weights (*p* < 0.01) were significant for age with calves born from older cows weighing more than younger cows. Weaning weight ratio displayed a linear increase with increasing intake classification (*p* < 0.01). Heifer yearling BW was significant for intake (*p* < 0.01) with low and average intake heifer classifications having greater heifer yearling BW than cows that had high intake classification as a heifer. Age and intake classification did not impact (*p* ≥ 0.22) pregnancy status or AI conception. In summary, heifer post-weaning feed intake classification had only minor impacts compared to age effects on lifetime productivity of Angus beef females.

## 1. Introduction

The profitability of beef cow-calf production systems is dependent on input and output costs [1]. Reducing input associated with developing replacement heifers can help reduce cost of production. Development and selection of replacement beef heifers is a key factor to improving overall herd productivity and profitability keeping in mind that feed costs represent about two-thirds of the cost of production for beef cattle producers [2]. In addition, the most common period to manipulate heifer growth and development is between weaning and the first breeding season [3]. Therefore, we need to consider the relationship between heifer development and heritability parameters to improve beef heifer growth and performance in beef cattle production systems. 

Moreover, variation in individual animal intake is impacted by stage of production, feeding behavior, environmental challenges, and the specific stage of the animal’s development [1]. Herd and colleagues [4] identified several physiological processes that impact feed intake across species, specifically digestion of feed, activity level, thermoregulation and metabolism. In order to maximize individual animal growth and overall productivity, we must match animal type to the environment. In low quality forage scenarios, ruminants with greater rumino-reticular capacity are more efficient [5] presumably due to the potential for greater intake and/or retention time. Heifer development programs must consider a synergistic combination of nutrition, genetics and environment in order to allow females to be the most productive over a lifetime. Therefore, it is important to consider potential metrics in beef heifer selection to improve lifetime productivity. 

When considering replacement heifers, growth traits (frame size and capacity) are two of the biggest considerations for beef producers as increased size and capacity allow for greater feed intake, as well as the ability to carry a calf. In relation, the expression of feed efficiency traits is dependent on the stage of maturity at which the trait was measured on the animals [1]. Parsons and coworkers [6] determined that residual feed intake (RFI) estimates on post-weaning heifers did not significantly predict subsequent lifetime productivity of the cow herd. In addition, Brody [7] indicated that puberty in heifers leads to changes in the growth curve of developing females which shifts from lean growth towards fat deposition, which could in essence impact intake due to physiological changes. However, having the ability to measure individual intakes on weaned heifers allows us to better understand the relationship between intake, size and potential lifetime productivity. We hypothesized that heifer post-weaning intake classification would not impact lifetime productivity and cow age would have a greater impact on lifetime productivity. Therefore, objectives of this study were to evaluate the relationship between post-weaning heifer intake (g/kg BW) classification on subsequent lifetime productivity of Black Angus beef females. 

## 2. Materials and Methods

All animals used in this study were housed at the Northern Agricultural Research Center (NARC), a part of the Montana Agricultural Experiment Station, (Havre, MT, USA; 48.5500° N, 109.6841° W). Experimental procedures utilized in this study were approved by the MSU Agriculture Animal Care and Use Committee of Montana State University. 

### 2.1. Feeding Trial and Baseline Data 

Comprehensive cow and calf performance data have been collected at NARC since 2010. Animals used in this study are commercial Black Angus females born between 2010 and 2019. Data on all females are collected starting at birth and ends when an animal is culled from the herd. To determine the impact of post-weaning voluntary feed intake, all NARC heifer calves are weaned onto pasture from late September to early October as a cohort. Approximately 60 to 85 d post-weaning, heifers were placed in pens equipped with GrowSafe (GrowSafe DAQ 4000E; GrowSafe System Ltd., Airdrie, AB, Canada) feed intake units on a feed intake trial where daily individual animal intake was collected. Heifers were provided primarily forage-based diet comprised of corn silage (55–74%) and grass hay (6.5–27%) on as fed basis. Individual feedstuff amounts varied from year to year based on availability, but all diets were formulated to meet the maintenance requirements for growing, moderate framed beef heifers (10.5% crude protein and 66.0% total digestible nutrients) [8]. To avoid over-conditioning, diets were calculated for 0.0 ADG when estimating 2% of BW intake; however, with unlimited intake the heifers usually gained 0.68 to 0.91 kgs per day [6]. Heifer BW was collected over two consecutive days at the start and end of the 80-d period, and every 28 d throughout the duration of the study in order to determine BW change over time. Heifers were categorized based on voluntary feed intake (g kg BW^−1^) as either low (*n* = 161; <−0.50 SD from the mean), average (*n* = 200; ±0.50 SD from the mean), or high (*n* = 158; >0.50 SD from the mean) within year. Cows were grouped according to their age into five categories, 2-year-olds, 3-year-olds, 4-year-olds, 5- and 6-year-olds and 7 years or older. Five- and six-year-old cows were combined to ensure group sizes were similar. 

### 2.2. Heifer Post-Intake Test Management 

Following the intake test, diets were reformulated to provide adequate nutrients for heifers to gain 0.68 kg/d and were held in a dry lot until turned out to pasture between April and May each year when range was available. Post-breeding, first-calf heifers and mature females were wintered together as a cohort at NARC headquarters and grazed summer pastures at the Thackeray Ranch (48°21′ N 109°30′ W), which is located south of Havre, MT in the Bear Paw Mountains. All females, first calf heifers and mature cows, were estrous synchronized and artificially inseminated based on timed AI in early June. Natural service bulls were presented to the females for an additional 45 d. Based on these time frames, females were expected to calve from March to April of the following year. At weaning, between mid-September and early October, all females were ultrasounded by a qualified veterinarian to determine pregnancy status. At weaning all calves were returned to the NARC hay grounds for grazing. All pregnant cows remained at the Thackeray Ranch through early January and grazed late season dormant forages. All females that were culled from the herd were recorded for cause of culling which included pregnancy status (open or out of normal breeding season), disposition concerns, or displayed structural concerns (lameness, teeth, feet/legs, udder).

### 2.3. Statistical Analysis 

Data utilized in this manuscript is located in Appendix A. Beef female production and reproduction data including cow BW and BCS at breeding, cow BW at weaning, calf birth weight, calf 205-d weight, calf weaning weight, calf weaning weight ratio (WWR) and Julian birth date were analyzed using ANOVA with a mixed model that included age, intake classification and the interaction of age and intake classification as fixed effects, and individual cow as the random effect (lme4; [9]). Data were plotted and transformed if needed to satisfy assumptions of normality and homogeneity of variance. Preplanned, orthogonal polynomial contrasts were used to determine linear and quadratic effects for intake and age main effects. Individual animal was considered the experimental unit. In addition, pregnancy status and AI conception rate were analyzed using generalized linear models following a binomial distribution in an ANOVA framework (car; [10]) (glm; [11]). An alpha ≤ 0.05 was considered significant. Tendencies were reported when significance was *p* ≤ 0.10. The Tukey method was used to separate means when alpha was <0.05 (emmeans; [12]). All statistical analyses were performed in R [11].

## 3. Results

For clarification, the results presented as cow are the heifer’s lifetime data from the initial RFI trial. The calf related data are from the offspring of the heifers in the initial RFI trial. These data were provided in order to determine if intake classification as a heifer impacted their subsequent production as a cow. There were no significant intake × age interactions (*p* ≥ 0.17; Table 1) for lifetime production variables. Therefore, only main effects will be presented.

### 3.1. Intake Classification Effects

Data related to the effect of heifer post-weaning intake classification (g/kg BW) on production and reproductive characteristics can be found in Table 1. Calf birth weight, 205-d weight, calf weaning weight, calf Julian birth date, and pregnancy status were not affected (*p* ≥ 0.15) by intake classification. Cow BW at breeding and weaning and cow BCS at breeding (*p* ≤ 0.04) displayed a linear decrease from females classified from low to high intake. Weaning weight ratio displayed a linear increase within increasing intake classification (*p* < 0.01). AI conception rates showed a tendency (*p* = 0.08) to increase linearly with increasing intake classification. 

Data related to the influence of post-weaning heifer intake classification (g/kg BW) on subsequent beef heifer yearling weight, total pounds of calf weaned and cow longevity spanning 5-breeding seasons and 4 weaned calf crops is in Table 2. Heifer yearling weight displayed a quadratic effect (*p* = 0.04) with heifers classified as low or average intakes having greater yearling weights than high intake classification females. Total pounds of calf weaned displayed a linear tendency for intake classification (*p* = 0.07) with low and average intake females weaning more total pounds over their lifetime compared to high intake classified females. Heifer intake classification displayed a linear tendency (*p* = 0.10) for percent of females remaining in the herd at 5-yrs with high intake classified females remaining in the herd longer than low and average intake classified females. 

### 3.2. Cow Age Effects

Data related to the effect of cow age on production characteristics in Angus beef females can be found in Table 3. Cow BW and BCS at breeding (*p* < 0.001, Table 3) displayed an age effect with cow BW increasing as the females aged. Body condition scores increased linearly from 5.08 in 2-year-old cows to 5.55 in five- and six-year-old cows, before decreasing to 5.33 in cows 7-years-old and greater. In addition, cow BW at weaning also showed significance for age (*p* < 0.001) and intake (*p* < 0.001) with younger cows being lighter than older cows, while low and average intake classified females were heavier than high intake classified females. Additionally, calf 205-d weaning weights (*p* < 0.01) and calf weaning weights (*p* < 0.01) were affected by cow age with calves born from older cows weighing more at 205-d and weaning when compared younger females. Calf Julian birth date, WWR, pregnancy status and AI conception were not influenced by cow age (*p* ≥ 0.15). 

Cow weights at breeding and weaning displayed a quadratic effect (*p* < 0.001) with weights at both timepoints increasing with increasing age. In addition, BCS also displayed a quadratic effect (*p* < 0.001) with increasing body condition as the cows aged. Calf data including calf birth weight, 205-d weights and weaning weights also displayed a quadratic effect (*p* < 0.001). As cows age, the weights of their calves at all three timepoints progressively increased with increasing age. In relationship, WWR displayed a quadratic tendency (*p* = 0.09) with younger cows having greater WWR values than older cows through six years of age. There was no linear or quadratic effect of Julian birth date, pregnancy status or AI conception, respectively.

## 4. Discussion

### 4.1. Cow Effects

Cow BW and BCS at breeding demonstrated that younger cows are lighter and have reduced BCS compared to mature cows, which was to be expected. Cassady and colleagues [13] investigated the relationship between heifer feed efficiency measures and intake of high- and low-quality forages and the impact on mature beef cows. Their results indicated that heifers that consumed less feed than the herd average had a 7% reduction in mature cow BW. In addition, cows that ate more feed as heifers, had the greatest DMI as cows, regardless of feed quality classification. Specifically, cows that consumed more feed/forage as heifers consumed greater than 17% more forage compared to cows that were classified as low or moderate intake as heifers. 

Consistent with current results, Turner and colleagues [14] reported that older cows weaned heavier calves compared to younger cows. In addition, calves that were born to older dams had greater calf birth weights, 205-d weights and weaning weights than calves that were born to younger cows. The heavier calves at birth may be due to the allotment of nutrients by the cow during pregnancy, as older cows no longer need to grow, allowing for more nutrients to be partitioned to the fetus. 

Data from Ziegler and coworkers [15] used crossbred mature beef cows, age 5 and over and determined that cow BCS at calving, pre-breeding and weaning were positively correlated with increased cow BW. For every 100-kg increase in cow BW, calf birth weight increased by 2.65 kg and adjusted 205-d weights were increased by 14.54 kg. Moreover, BW of heifer progeny post-weaning increased with every additional 100-kg increase in dam BW. However, in the current study, BCS did increase as BW increased as cows aged, but this increase in BCS plateaued at 5–6 years of age. When compared to the current results, a similar increase was observed with cow BW at breeding and weaning between the 2- and 5-yr old cows being approximately 100-kg difference. In the current study, calf birth weight and 205-d weight increased approximately 16 and 14%, respectively in the heavier 5-yr old cows compared to the 2-yr old cows. These weight differences may be related to the utilization of calving ease bulls on 2-yr-olds versus mature cow bulls on the older females. 

Cows conceiving to AI did not differ by intake classification with AI conception averaging 62% across all intake classifications. This is likely due to the fact that all females, regardless of age had sufficient body condition in order to successfully conceive. Additionally, there was no difference in pregnancy status based on intake classification. Therefore, heifer intake classification may have minimal impacts on reproductive performance. 

Data from Stewart and Martin [16] reported that smaller framed Angus cows remained in the herd longer than larger framed Angus cows, therefore producing more calves over their lifetime. The current study observed that herd longevity tended to increase linearly from low to high intake designation. In general, it is expected that high intake cows will be large framed, which would indicate that the current study suggests herd longevity is opposite of Stewart and Martin [16]. However, as discussed previously, when classifying intake on a per kg BW basis, the classification is greatly influenced by BW and in addition to the lack of influence on calf production, the larger framed/heavier cows may be in the low intake group. 

The use of intake (body weight basis) to select heifers has not been readily explored in the literature due to the practical challenges in acquiring individual animal intake estimates. In low-quality, high-fiber forage systems, the ability to select animals with greater ruminal-reticular capacity may yield cattle that are more efficient because of the greater intake potential and the possibility for increased ruminal retention/digestion of low-quality forages [5]. This does not suggest that the best animals are the biggest with the largest ruminal-reticular volume. Rather, the most efficient animals are those with the greatest portion of their body weight consisting of the ruminal-reticular column and content. In our study, the post-weaning heifer intake classification was influenced by heifer weight. Specifically, the lighter heifers were the individuals more likely to have high intake per unit body weight. This observation is supported by intake models provided by the Beef Cattle NASEM [8] for growing animals which suggest the intake of weaned animals is greater than yearling intakes when expressed on a body weight basis. As a result, using intake (body weight basis) as a metric for heifer selection may be biased by heifer physiological maturity, and metabolic status. 

### 4.2. Calf Effects

There were no significant impacts of intake classification on calf data except for WWR. These data indicated that high intake classified cows weaned heavier calves than low and average intake classified cows. This difference may be due to increased rumino-reticular volume, leading to increased intake and therefore potential for increased milk production. As expected, birth weights were lighter for calves born to younger cows than older cows, likely due to the selection of calving ease bulls on younger females. This is consistent with data from Beard and colleagues [17], who reported that heifer calves that were born to young cows had lighter birth weights than those born to moderate (4 to 6 yr old) and old cows (≥7 yr old). Contrary to the current results, Beard and colleagues [17] reported that calves born to moderate aged or old cows had a greater percentage of offspring that reached puberty prior to their first breeding season when compared to heifers born to young cows. 

## 5. Conclusions

Classification of heifers based on post-weaning intake classifications had very little effect on subsequent lifetime productivity measures. Slight differences were displayed in heifer BWs with heifers classified as low intake being greater BW than those classified as average or high intake. However, low intake classified females were also culled from the herd sooner than females that were classified as average or high intake. Therefore, this study suggests the selection of replacement heifers based on post-weaning intake classification following weaning may not be the best point in the production cycle as younger/smaller heifers may be at a disadvantage compared to their larger/older counterparts. Therefore, further research is necessary to determine when to collect post-weaning intake information to provide the best indication of subsequent lifetime productivity of replacement females.

## Figures and Tables

**Table 1 animals-12-01687-t001:** The influence of heifer post-weaning intake classification expressed as g per kg body weight on subsequent production and reproductive characteristics in Angus beef females.

Category	Intake Classification ^1^			SE ^2^	*p*-Value			Contrasts	
	Low	Average	High		Intake	Age	Intake × Age	Linear	Quadratic
Cow BW at breeding, kg	571.32	565.29	550.13	11.37	0.15	<0.001	0.92	0.003	0.41
Cow BCS at breeding	5.38	5.33	5.29	0.08	0.58	<0.001	0.99	0.04	0.96
Cow BW at weaning, kg	611.97 ^a^	606.49 ^a^	587.87 ^b^	6.39	<0.001	<0.001	0.67	<0.001	0.20
Calf birth weight, kg	40.79	39.95	40.42	0.75	0.37	<0.01	0.32	0.52	0.14
Calf weaning wt., kg	251.94	250.20	252.65	6.85	0.27	<0.01	0.45	0.79	0.30
205 d wt., kg	270.97	268.72	271.04	6.58	0.63	<0.01	0.34	0.98	0.27
Calf Julian birth date	83.40	83.20	82.88	1.75	0.49	0.15	0.40	0.68	0.95
Weaning weight ratio ^3^	41.37 ^a^	41.65 ^a^	43.38 ^b^	1.24	<0.01	0.67	0.17	<0.01	0.15
Pregnancy Status, %	88.26	88.32	86.51	1.51	0.22	0.65	0.38	0.45	0.60
AI Conception, %	57.77	63.46	64.70	2.61	0.38	0.80	0.63	0.08	0.47

^1^ Heifers were categorized as either low (>−0.50 SD from the mean; *n* = 161), or average (±0.50 SD from mean; *n* = 904) or high (<+0.50 SD from the mean; *n* = 158) intake classes. ^2^ Pooled standard error of the mean. ^3^ Weaning weight ratio = calf 205-d actual weaning wt./cow wt at weaning. ^a,b^ Means within row and cow age lacking common superscript differ (*p* < 0.05).

**Table 2 animals-12-01687-t002:** The influence of heifer post-weaning intake expressed as grams per kg of body weight on subsequent heifer yearling weight, total lbs. of calf weaned and cow longevity spanning 5-breeding seasons and 4 weaned calf crops.

Category	Intake Classification ^1^			SE ^2^	*p*-Value		
	Low	Average	High		Intake	Linear	Quadratic
Heifer yearling BW, kg	380.92 ^a^	376.05 ^a^	358.01 ^b^	7.00	<0.01	<0.001	0.04
Total lbs. of calf weaned, kg ^3^	484.46	487.25	415.50	27.17	0.09	0.07	0.25
Present at 5 yr, %	63.22	72.12	74.73	4.71	0.22	0.10	0.62

^1^ Heifers were categorized as either low (>−0.50 SD from the mean; *n* = 161), or average (±0.50 SD from mean; *n* = 200) or high (<+0.50 SD from the mean; *n* = 158) intake classes. ^2^ Pooled standard error of the means. ^3^ Limited to cows that had the opportunity to have at least four calves and were bred for the fifth. ^a,b^ Means within the row that lack common superscripts differ (*p* < 0.05).

**Table 3 animals-12-01687-t003:** The influence of cow age on production and longevity characteristics in Angus beef females.

Category	Cow Age, Years					SE ^1^	*p*-Value		
	2	3	4	5–6	7+		Intake	Age	Intake × Age
Cows, n	441	349	278	356	258				
Cow wt. at breeding, kg	485.08 ^a^	528.58 ^b^	579.89 ^c^	600.83 ^d^	616.87 ^d^	11.42	0.15	<0.001	0.92
Cow BCS at breeding	5.08 ^a^	5.26 ^b^	5.44 ^c^	5.55 ^d^	5.33 ^b,c^	0.08	0.58	<0.001	0.99
Cow wt. at weaning, kg	522.90 ^a^	572.37 ^b^	620.13 ^c^	645.18 ^d^	649.96 ^d^	6.13	<0.001	<0.001	0.67
Calf birth weight, kg	34.19 ^a^	39.77 ^b^	41.44 ^c^	42.78 ^c,d^	43.77 ^d^	0.77	0.32	0.52	0.14
Calf weaning wt., kg	224.04 ^a^	244.53 ^b^	256.10 ^c^	261.87 ^c^	271.43 ^d^	6.86	0.27	<0.01	0.45
205 d wt., kg	238.27 ^a^	262.37 ^b^	275.65 ^c^	281.81 ^d^	293.12 ^e^	6.59	0.63	<0.01	0.34
Calf Julian birth date	81.53	83.55	83.54	83.31	83.87	1.83	0.49	0.15	0.40
Weaning rate ratio ^2^	42.77	42.53	41.84	41.41	42.12	1.23	<0.01	0.67	0.17
Pregnancy status, %	88.90	87.85	87.96	85.28	88.36	1.93	0.22	0.65	0.38
AI Conception, %	58.49	58.74	65.19	61.19	66.22	3.31	0.38	0.80	0.63

^1^ Pooled standard error of the means. ^2^ WWR = calf 205-d actual weaning wt./cow wt at weaning. ^a–e^ Means within row lacking common superscript differ (*p* < 0.05).

## Data Availability

Data are available in Appendix A.

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
