# Peer review of "Influence of Heifer Post-Weaning Voluntary Feed Intake Classification on Lifetime Productivity in Black Angus Beef Females"

_animals, 2022, doi:10.3390/ani12131687_

Round 1

Reviewer 1 Report

General Comments:

Overall, the manuscript reads well, as it is concise and well structured. The authors did a great job of building the manuscript with the reader in mind. Plenty of replicates with year and the number of head used for the experiment, so a strong data set. A few comments would be to comb over the manuscript for abbreviations and make sure they’re introduced prior to use, highlighted a few in the minor comments below. Clarification points would be to highlight how the herd is managed, such as do the 2-3 year olds calve out earlier than the mature herd or are they all run together? Could influence post-natal performance of those heifer calves altering growth trajectory. Would also consider looking into some of the suggested reproductive traits below to strengthen the reproductive performance impact.

Line 28: define BW

Line 63: sentence flow is off, currently it reads as highlighting selection pressure for heifer growth traits, yet authors are saying it’s two of the biggest considerations. Consider rewording the sentence to highlight the intended focus on frame size and capacity. On that note maybe add a sentence or two about genetic lineage. Many producers will look to dam lines or sire lines as well not just individual heifer size.

Line 66: define RFI

Line 73: No indication of a hypothesis, only objective is outlined.

Line 85: Are pastures homogeneous? Are all heifer calves ran as a cohort? Could have an impact on heifer performance during the transition into GrowSafe.

Line 89: if authors have the data, would be nice to see a table of a proximate analysis of the average diet fed. 

Line 104: I know this is post-intake, but what is the rationale for 0.68 kg/d of gain? How does this relate to the typical 50 -65% rate of gain target in developing heifers with reference to your herd? 

Line 106: Does beef female just mean the beef cows and heifers? Would fix throughout document.

Line 152: if the authors have the data, due to importance of heifer selection and expectation for the herd it would be beneficial to evaluate intake classification on puberty attainment prior to the breeding season and calving within the first 21 days of the season. (supporting papers from Cushman, Summers, and Moriel labs) 

Line 184: add WWR in parentheses after weaning weight ratio

Author Response

General Comments:

Overall, the manuscript reads well, as it is concise and well structured. The authors did a great job of building the manuscript with the reader in mind. Plenty of replicates with year and the number of head used for the experiment, so a strong data set. A few comments would be to comb over the manuscript for abbreviations and make sure they’re introduced prior to use, highlighted a few in the minor comments below.

Clarification points would be to highlight how the herd is managed, such as do the 2-3 year olds calve out earlier than the mature herd or are they all run together? Could influence post-natal performance of those heifer calves altering growth trajectory. Would also consider looking into some of the suggested reproductive traits below to strengthen the reproductive performance impact.

Author response: general herd management including all cows is presented in lines 109-124

Line 28: define BW 

Author response: changed as suggested

Line 63: sentence flow is off, currently it reads as highlighting selection pressure for heifer growth traits, yet authors are saying it’s two of the biggest considerations. Consider rewording the sentence to highlight the intended focus on frame size and capacity. On that note maybe add a sentence or two about genetic lineage. Many producers will look to dam lines or sire lines as well not just individual heifer size.

Author response: reworded for clarification

Line 66: define RFI

Author response: changed as suggested

Line 73: No indication of a hypothesis, only objective is outlined.

Author response: hypothesis has been included

Line 85: Are pastures homogeneous? Are all heifer calves ran as a cohort? Could have an impact on heifer performance during the transition into GrowSafe.

Author response: more information provided for clarification

Line 89: if authors have the data, would be nice to see a table of a proximate analysis of the average diet fed. 

Author response: due to the variability of the diet each year we do not have a table with proximate analysis. We included additional information into the text in line 96-97

Line 104: I know this is post-intake, but what is the rationale for 0.68 kg/d of gain? How does this relate to the typical 50 -65% rate of gain target in developing heifers with reference to your herd? 

Author response: The 0.68 kg/d was to ensure adequate weight gain for developing heifers to reach 50-65% of mature weight for breeding.

Line 106: Does beef female just mean the beef cows and heifers? Would fix throughout document.

Author response: reworded for better understanding

Line 152: if the authors have the data, due to importance of heifer selection and expectation for the herd it would be beneficial to evaluate intake classification on puberty attainment prior to the breeding season and calving within the first 21 days of the season. (supporting papers from Cushman, Summers, and Moriel labs) 

Author response: no data was collected to evaluate puberty attainment prior to breeding

Line 184: add WWR in parentheses after weaning weight ratio

Author response: Weaning weight ratio was previously defined in line 128 and is not needed to be defined again

Reviewer 2 Report

Line 23 – it feels repetitive to include (g/kg BW) after VFI each time, particularly without any values

Line 26 – should this be >0.05? to be consistent with the other parameters?

Line 33 – Use cow yearling and heifer in the same sentence for what I assume is describing the same group of animals. Please be more consistent with animal terms to make it clearer what/who is being described.

Line 45 – “feed costs”

Line 49 – This line seems unnecessary. Perhaps there’s debate on how to accomplish these goals, but not that it is expensive or a lengthy time process. Recommend removing or better wording.

Line 56 – change ; to ,

Line 66 – please define RFI

Line 73 – since you are not giving a value, the units are unnecessary

Line 93 – define 0.0 ADG – it appears you calculated the diets for animals not to gain weight post weaning. More context to this term would be useful.

Line 96 – is this every 28 d until animals are culled? How long were animals weighed for?

Line 98 – In the abstract, you indicated 2,204 heifers were included within the study. Here you have n = 519 across treatments. Please make animal numbers consistent.

Line 99 – Same comment about 0.50 vs 0.05 SD

Line 106 – “Beef females, first-calf heifers, and mature females” is confusing. Who are the beef females if not the other two listed? Bred heifers?

Line 122 – change included to including

Line 125 – was age alone used as a fixed effect?

Line 120 – I’m struggling to understand your actual experimental design and true experimental unit. Whose post weaning intake is being categorized? Is it the cow and you’re following her calf performance over her calvings until culling to determine if her previous intake impacts her calf’s performance? Or are you following the calf and it’s post weaning intake and including performance metrics of the cow? Since the terms calf, heifer, and cow are used so consistently when discussing both groups of animals, it would be helpful to better understand which group is being discussed as the experimental unit with more defined terms.

Line 191 – While this discussion is interesting, it is about feed efficiency and not about the result of animal age on weight and BCS. It is not supporting or discussing the result.

 Line 224 – remove cows before Angus

Author Response

Line 23 – it feels repetitive to include (g/kg BW) after VFI each time, particularly without any values

Author response: the authors use g/kg BW after each VFI to clarify the measurement on a per kg BW basis, and not on the assumption that would be total kg intake

Line 26 – should this be >0.05? to be consistent with the other parameters?

Author response: has been made consistent

Line 33 – Use cow yearling and heifer in the same sentence for what I assume is describing the same group of animals. Please be more consistent with animal terms to make it clearer what/who is being described.

Author response: changed to better describe the effect

Line 45 – “feed costs”

Author response: changed as suggested

Line 49 – This line seems unnecessary. Perhaps there’s debate on how to accomplish these goals, but not that it is expensive or a lengthy time process. Recommend removing or better wording.

Author response: removed the sentence

Line 56 – change ; to ,

Author response: changed as suggested

Line 66 – please define RFI

Author response: changed as suggested

Line 73 – since you are not giving a value, the units are unnecessary 

Author response: changed as suggested

Line 93 – define 0.0 ADG – it appears you calculated the diets for animals not to gain weight post weaning. More context to this term would be useful.

Author response: added extra content for clarification

Line 96 – is this every 28 d until animals are culled? How long were animals weighed for? further clarification provided

Author response: sentence modified to add clarification

Line 98 – In the abstract, you indicated 2,204 heifers were included within the study. Here you have n = 519 across treatments. Please make animal numbers consistent.

Author response: has been addressed

Line 99 – Same comment about 0.50 vs 0.05 SD

Author response: has been address

Line 106 – “Beef females, first-calf heifers, and mature females” is confusing. Who are the beef females if not the other two listed? Bred heifers?

Author response: sentence restructured for clarification

Line 122 – change included to including

Author response: changed as suggested

Line 125 – was age alone used as a fixed effect?

Author response: Clarification was included.

Line 120 – I’m struggling to understand your actual experimental design and true experimental unit. Whose post weaning intake is being categorized? Is it the cow and you’re following her calf performance over her calvings until culling to determine if her previous intake impacts her calf’s performance? Or are you following the calf and it’s post weaning intake and including performance metrics of the cow? Since the terms calf, heifer, and cow are used so consistently when discussing both groups of animals, it would be helpful to better understand which group is being discussed as the experimental unit with more defined terms.

Author response: individual animal served as the experimental unit. Heifers born between 2010 and 2019 were used in this experiment and were categorized based on their voluntary feed intake during the RFI period. Individual intake was measured, so individual heifer was the experiment unit for intake classification. The data presented as “cow” are the heifers from the RFI trial. The data presented were to determine if intake classification as a heifer impacted their subsequent production as a cow. The “calf” data are the calves from those cows from the heifer RFI trial as they age. Additional text was included to clarify

Line 191 – While this discussion is interesting, it is about feed efficiency and not about the result of animal age on weight and BCS. It is not supporting or discussing the result.

Author response: little information is available for lifetime productivity based on heifer feed efficiency. The little information available is also limited in its scope of data. Also, based on our results, age had a larger impact on lifetime productivity than feed intake, therefore, we discussed this and there is more information available.

Line 224 – remove cows before Angus

Author response: changed as suggested